# Online System for Monitoring the Degree of Fermentation of Oolong Tea Using Integrated Visible–Near-Infrared Spectroscopy and Image-Processing Technologies

**DOI:** 10.3390/foods13111708

**Published:** 2024-05-29

**Authors:** Pengfei Zheng, Selorm Yao-Say Solomon Adade, Yanna Rong, Songguang Zhao, Zhang Han, Yuting Gong, Xuanyu Chen, Jinghao Yu, Chunchi Huang, Hao Lin

**Affiliations:** 1School of Food and Bioengineering, Jiangsu University, Zhenjiang 212013, China; 17853260667@163.com (P.Z.); syadade@gmail.com (S.Y.-S.S.A.); wangjun18763606962@163.com (Y.R.); zsgemail9679@163.com (S.Z.); 15695117976@163.com (Z.H.); swchenxuanyu@gmail.com (X.C.); yujinghao878@163.com (J.Y.); 2Chichun Machinery (Xiamen) Co., Ltd., Xiamen 361100, China; 17553735899@163.com (Y.G.); lzy9854@163.com (C.H.)

**Keywords:** Oolong tea, fermentation, data fusion, machine learning, prediction

## Abstract

During the fermentation process of Oolong tea, significant changes occur in both its external characteristics and its internal components. This study aims to determine the fermentation degree of Oolong tea using visible–near–infrared spectroscopy (vis-VIS-NIR) and image processing. The preprocessed vis-VIS-NIR spectral data are fused with image features after sequential projection algorithm (SPA) feature selection. Subsequently, traditional machine learning and deep learning classification models are compared, with the support vector machine (SVM) and convolutional neural network (CNN) models yielding the highest prediction rates among traditional machine learning models and deep learning models with 97.14% and 95.15% in the prediction set, respectively. The results indicate that VIS-NIR combined with image processing possesses the capability for rapid non-destructive online determination of the fermentation degree of Oolong tea. Additionally, the predictive rate of traditional machine learning models exceeds that of deep learning models in this study. This study provides a theoretical basis for the fermentation of Oolong tea.

## 1. Introduction

Oolong tea is one of the six major tea types in China, belonging to the category of semi-fermented teas, and is known for its mellowness, as well as its strong fruity and floral aroma. Oolong tea is primarily cultivated in Fujian, Guangdong, and Taiwan Provinces [1]. Among these, Fujian Province holds a significant position, contributing to 75% of the total national production, solidifying its reputation as the quintessential province for Oolong tea production [2]. Oolong tea is a high-quality tea produced through a series of processes including picking, withering, fermentation, and drying. The fermentation process is a crucial stage for developing the distinctive quality of Oolong tea due to the external characteristics and internal components being changed significantly [3]. This transformation is accompanied by the generation of aromatic compounds, contributing to the development of a nuanced floral and fruity aroma [4,5]. Therefore, controlling the fermentation process has a crucial role in monitoring the quality of tea. Insufficient or excessive fermentation can significantly impact the quality of Oolong tea. In the traditional tea-making process, the fermentation level of tea leaves is typically judged by relying on the experience of the craftspeople [6]. However, this subjective assessment is not conducive to industrial production. As a result, economical and rapid identification methods are needed.

To objectively evaluate the fermentation level of Oolong tea, various non-destructive testing technologies, such as electronic nose [7] and hyperspectral [8] techniques, have been applied. Nevertheless, these techniques are complex, time-consuming, and susceptible to environmental factors. Therefore, the rapid, convenient, efficient, and accurate assessment of Oolong tea fermentation remains an unresolved issue. Visible–near-infrared spectroscopy (vis-VIS-NIR) technology is a non-destructive technique capable of providing information about the chemical composition of a sample, including its organic compounds and moisture content, without altering the sample itself. This capability enables VIS-NIR to rapidly and accurately assess changes in the internal components of tea leaves during the fermentation process of Oolong tea [9]. Furthermore, computer vision technology enables the acquisition of visual information from samples, including information about their appearance, color, shape, and surface features. It also facilitates real-time monitoring by integrating various image features for multidimensional analyses. Therefore, computer vision technology can precisely capture external color and morphological changes in objects [10]. The fermentation process of Oolong tea involves numerous physicochemical changes [11], and it is challenging for a single sensor to assess the intricate variations occurring during Oolong tea fermentation. Therefore, the accuracy of assessment can be enhanced through data fusion [12]. Both VIS-NIR and computer vision technology can be employed in conjunction with variable-selection algorithms and data fusion to rapidly and conveniently evaluate the fermentation process of Oolong tea. Currently, there is no research that has reported on the application of data fusion technology in the Oolong tea fermentation process.

Therefore, our research focuses on the use of VIS-NIR and computer vision combined with data fusion to evaluate the quality of Oolong tea during fermentation. This study is divided into the following segments: (I) The collection of near-infrared spectroscopy and image information during the fermentation process of Oolong tea. (II) The extraction of feature variables from both near-infrared spectroscopy and image information, followed by data fusion. (III) The development of traditional machine learning models and deep learning models using data fusion to classify the degree of fermentation of Oolong tea.

## 2. Materials and Methods

### 2.1. Fermentation Materials and Sample Collection

The variety of Oolong tea used in this study was Iron Goddess Tea, harvested according to the standard of picking one bud and two leaves from two-year-old tea bushes. The leaves were sourced from the Tieguanyin plantation base in Longjuan Township, Anxi County, Quanzhou City, Fujian Province. The experiment was conducted on 25 August 2022.

First, 100 kg of freshly picked tea leaves were spread out to a thickness of 5 cm and left at room temperature (25 °C) for 2 h. Subsequently, they were placed into a comprehensive fermentation machine (manufactured by Chichun Machinery, Xiamen Co., Ltd., Xiamen, China) for fermentation. The fermentation machine was set at a temperature of 32 °C with a drum rotation speed of 30 rotations per minute. After approximately 2 h of rolling, the machine was stopped, and the leaves were left for natural fermentation for 3.5 h. In order to investigate a situation of excessive fermentation, the fermentation process was extended to 7.5 h.

According to the experience and traditional habits of tea makers, the first 5.5 h after the beginning of fermentation is the underfermentation stage, 5.5 to 6.5 h after the start of fermentation is the moderate fermentation stage, and 6.5 to 7.5 h after the beginning of fermentation is the excessive fermentation stage. This study divided the degrees of fermentation according to this traditional scale. Samples were collected every ten minutes during the fermentation process; 7 samples were collected at a time. Each sample weighed approximately 50 g, and sampling was performed 45 times, yielding a total of 315 samples throughout the process.

### 2.2. Device Requirement

In order to predict the fermentation degree of Oolong tea successfully, we needed to obtain the near-infrared spectral data and image data during the fermentation process of Oolong tea. However, due to the complex shape and uneven distribution of Oolong tea in the fermentation process, it was inconvenient to collect the spectral data of Oolong tea during fermentation. Therefore, it was necessary to employ a quartz plate to uniformly compress the tea leaves; however, manual compression may introduce human-induced errors, thereby impacting the experimental outcomes. In addition, if manual detection is used, the detection process is not intelligent enough, and the detection efficiency is not high, which will lead to a prolonged detection cycle, resulting in experimental errors. In order to solve the above problems and obtain accurate and stable experimental data during the fermentation of the Oolong tea, this study developed an intelligent detection equipment to standardize the method of obtaining data.

### 2.3. Acquisition of Spectral and Image Signals

The tea fell evenly on the conveyor belt due to gravity. The conveyor belt transported the tea-leaf samples to the dark box of the image acquisition system with dimensions of 180 × 180 × 200 mm. The annular light source (from Dongguan, China Polyphoto Electric Co., Ltd., Dongguan, China) was activated, and a CCD camera (MER2-1220-32U3C-L, from Beijing China Daheng Group Inc., Beijing, China, with a resolution of 4024 × 3036) was positioned 120 mm away from the tea leaves for capturing images. One image was collected for each sample, resulting in the acquisition of 315 images from 45 batches.

After image acquisition, the conveyor belt continued to transport the tea-leaf samples to the dark box of the VIS-NIR acquisition system, with dimensions of 180 × 180 × 200 mm. Inside the dark box, two halogen tungsten lamps were turned on, and a small stepper motor connected to a ball screw (FSK40, from China Fuyu Technology Co., Ltd., Chengdu, China) drove the quartz plate to press the tea leaves on the conveyor belt. The reflected light passed through a fiber collimator (SMA905-FC-74UV, from China Wenyi Photoelectric Co., Ltd., Shenzhen, China) and optical fiber, which was received by the spectrometer, and the spectrometer (XS11639, from China Ruhai Photoelectric Technology Co., Ltd., Shanghai, China, with a spectral range of 190–1100 nm) was activated. In the reflection mode, spectral information of the tea leaves was collected with the integration time set to 30 ms, averaging 4 times, with pixel smoothing occurring 9 times. To reduce errors caused by environmental factors, 5 spectra were collected for each sample and then averaged, resulting in a total of 315 spectra from the 45 batches.

### 2.4. Image Feature Extraction

Image feature extraction involves the extraction of color features and texture features. In terms of color features, the values of R, G, B, H, V, S, L*, a*, and b* in three color spaces (RGB, HSV, Lab) of each image were extracted. The color range of the image was visualized, and an HSV color range 3D scatter plot was constructed. Based on the scatter plot, the color range for red was selected, and the area of the red range was calculated as a percentage S* of the total image area.

Concerning the texture features, we utilized a gray-level co-occurrence matrix to extract the texture features [13]. We used m (mean), v (variance), s (skewness), k (kurtosis), and e (entropy) to describe the texture information of the tea leaves. The mean described the overall grayscale distribution of the tea-leaf texture; a higher mean indicated a more uniform grayscale distribution of texture structures in the image. Variance described the local grayscale correlation on the surface of the tea leaves; a higher variance indicated significant differences in the grayscale among texture regions in the image. Skewness described the asymmetry of the grayscale distribution. Kurtosis described the sharpness of the grayscale distribution; higher kurtosis implied a sharp and concentrated grayscale distribution. Entropy described the level of chaos in the information in the tea-leaf images; a higher entropy value indicated a more irregular texture in the image.

### 2.5. Spectral Feature Extraction

#### 2.5.1. Spectral Preprocessing

In order to mitigate the impact of baseline shifts, noise, and stray light on the spectral data [14], this study employed two preprocessing methods: standard normal variate transformation (SNV) and Savitzky–Golay (SG) smoothing. The spectral range selected for analysis was 400–950 nm, resulting in a total of 1517 wavelength points.

#### 2.5.2. Spectral Feature Screening

This study involved the collection of VIS-NIR data, encompassing 1517 wavelength points, which can be regarded as high-dimensional data. Typically, spectral data include abundant extraneous information and noise variables [15]. Therefore, in our qualitative analysis using VIS-NIR, it was crucial to enhance the predictive accuracy by filtering feature wavelengths and establishing classification recognition models with higher predictive capabilities [16]. This research employed competitive adaptive reweighted sampling (CARS), variable combination population analysis (VCPA), variable combination population analysis with iteratively retaining informative variables (VCPA-IRIV), and a sequential projection algorithm (SPA) to select effective feature wavelengths for improving the predictive performance of the model.

The characteristics of these four variable-selection algorithms are outlined as follows:

CARS: CARS is a significant technique in the field of compressive sensing. It aims to overcome challenges posed by noise and limited observations to better reconstruct sparse signals. This algorithm performs well when dealing with highly sparse signals [17].

VCPA: VCPA helps to reduce redundancy in data, providing a more compact and representative set of variables. This, in turn, enhances modeling efficiency and effectiveness [18].

VCPA-IRIV: IRIV attempts to identify and retain variables crucial for model performance in a specific task. Combining VCPA with IRIV synergistically enhances the extraction and retention of key information during the feature selection and variable screening processes [19].

SPA: An SPA sequentially introduces new features, incorporating the most representative ones each time. This stepwise construction aids in dimensionality reduction while retaining crucial information, thereby improving the model’s generalization ability and interpretability [20].

These wavelength-screening algorithms have been successfully applied to screen features of VIS-NIR [21] and Raman spectra [22].

### 2.6. Data Fusion

In order to achieve an enhanced predictive model [23], this study integrated feature-level data fusion of variable-selected VIS-NIR information with image data. The emphasis in feature-level fusion is on extracting feature-variable data from different data sources and combining them into a new array to create the optimal model. Consequently, we further integrated the image data with the VIS-NIR spectral data selected using the four filtering algorithms. Theoretically, a feature-level fusion strategy can achieve information compression and improve efficiency while maximizing the retention of the original information in the data.

### 2.7. Identification Models for Determining the Fermentation Degree of Oolong Tea

#### 2.7.1. Three Traditional Machine Learning Models

This research employed linear discriminant analysis (LDA), support vector machine (SVM), and backpropagation artificial neural network (BPANN) algorithms for the classification and identification of the three fermentation stages of Oolong tea. 

The following are the characteristics of these three traditional machine learning algorithms:

***LDA:*** LDA is a supervised learning algorithm primarily used for dimensionality reduction and classification tasks. The objective of LDA is to find a hyperplane that best separates different classes, thereby maximizing the differences between the classes in a low-dimensional space while minimizing the differences within classes [24].

***SVM:*** An SVM is a commonly used supervised learning algorithm primarily for binary classification problems, though it can be extended for multi-class and regression tasks. It finds an optimal hyperplane in the feature space that maximizes the margin between samples of different classes, thus achieving classification [25].

***BPANN:*** A BPANN is an algorithm based on artificial neural networks (ANNs), which adjusts the network parameters during training by calculating errors and backpropagating to minimize the loss function, thus achieving the approximation of complex functions and pattern recognition [26].

The effectiveness of a classification model is evaluated using classification performance metrics based on the percentage of actual classified samples [27]. Performance parameters like sensitivity, specificity, accuracy, and the error rate are commonly obtained from the confusion matrix to evaluate classification performance. Sensitivity is the model’s ability to correctly identify the sample of the category under consideration; specificity is defined as the ability of the model to correctly reject all other classes of samples. Accuracy and the error rate refer to the classification model’s capacity to correctly identify or reject all samples, respectively.

#### 2.7.2. Two Deep Learning Models

Deep learning models are machine learning models that attempt to mimic the human brain. They focus on using neural networks to solve complex pattern-recognition problems. The characteristic of deep learning is that the model consists of multiple layers of non-linear processing units, which can automatically learn features from data. In this study, two neural networks, namely a convolutional neural network (CNN) and a multi-layer perceptron (MLP) model, were used to classify and recognize the fermentation degree of the Oolong tea, and were compared with the three traditional machine learning models (Figure 1).

The idea behind convolutional classification primarily leverages the characteristics of CNNs, which can capture local correlations, thereby transforming one-dimensional spectral data into high-dimensional features suitable for classification. In this process, convolutional layers and pooling layers play important roles. Convolutional layers extract features from input data using their filters, while pooling layers are responsible for reducing the dimensionality of the data, thereby reducing computation and preventing overfitting [28].

The MLP model, also known as an artificial neural network, consists of input, hidden, and output layers, and possesses good non-linear mapping capabilities. The approach an MLP takes for classification and recognition is through a feedforward neural network, where input data are passed through a network structure containing multiple hidden layers and activation functions. Neurons in each hidden layer extract abstract representations of input features by learning weight parameters, and classification is ultimately performed through the output layer. During training, the backpropagation algorithm adjusts network weights by minimizing the loss function, enabling the model to accurately predict the category of input data [29].

#### 2.7.3. Model Evaluation Index

When evaluating the recognition performance of discriminative models, it is essential to assess not only their predictive capabilities but also the nature of the errors in the model’s recognition results. Therefore, it is common to utilize classification performance metrics based on the actual percentage of classified samples to evaluate the effectiveness of classification models. Typically, performance parameters such as sensitivity, specificity, accuracy, and the error rate are derived from the confusion matrix to evaluate the classification performance. Sensitivity measures the model’s ability to correctly identify samples belonging to the considered class, while specificity measures the model’s ability to correctly reject all other class samples. Accuracy and the error rate refer to the model’s ability to correctly identify all samples or reject all samples, respectively. These parameters are calculated based on the true positives (TPs, correctly identified), true negatives (TNs, correctly rejected), false positives (FPs, incorrectly identified), and false negatives (FNs, incorrectly rejected) obtained during calibration validation, as computed using the following equations (Equations (1)–(4)):(1)$$\mathrm{Sensitivity}=\frac{\mathrm{TP}}{\mathrm{TP}+\mathrm{FN}}$$
(2)$$\mathrm{Specificity}=\frac{\mathrm{TN}}{\mathrm{TN}+\mathrm{FP}}$$
(3)$$\mathrm{Accuracy}=\frac{\mathrm{TP}+\mathrm{TN}}{\mathrm{TP}+\mathrm{TN}+\mathrm{FP}+\mathrm{FN}}$$
(4)$$\mathrm{Error}=1-\mathrm{Accuracy}=\frac{\mathrm{FP}+\mathrm{FN}}{\mathrm{TP}+\mathrm{TN}+\mathrm{FP}+\mathrm{FN}}$$

## 3. Results and Discussions

### 3.1. Device Construction

The monitoring system consisted of a dark box, a baffle, a conveyor belt, an image acquisition system, and a VIS-NIR acquisition system. The image acquisition system and VIS-NIR acquisition system were positioned within the dark box and separated by the baffle. The conveyor belt initially transports the tea leaves to the image acquisition system, where the light source is activated to capture image information. Subsequently, the tea leaves are conveyed to the visible–near-infrared (vis-VIS-NIR) spectroscopy acquisition system. Here, the quartz plate automatically descends to press the tea leaves firmly, the halogen tungsten lamp is activated, and near-infrared spectroscopy data are collected. Upon completion of data collection, the tea leaves are transported outside the monitoring system. Figure 2 shows a schematic diagram of the sampling process. This monitoring system was installed within the comprehensive fermentation machine produced by Chichun Machinery (Xiamen, Fujian Province, China), enabling in-process sampling analysis to achieve real-time monitoring of the Oolong tea fermentation levels.

### 3.2. Image Feature Extraction Results

With the increase in fermentation time, significant changes occurred in the color of the Oolong tea. As depicted in Figure 3a,b, the area of the red part of the leaves continuously grew from below 1% to around 20%, indicating a decreased trend in brightness (V and L) and saturation (S). This implied an overall reduction in image brightness and a deepening of the color. This phenomenon was attributed to friction between the leaves during fermentation, causing damage to the edge cells of the leaves. Polyphenol oxidases in the tea leaves reacted with oxygen in the air, leading to the oxidation of polyphenolic substances in the tea leaves and the formation of oxidized enzymes. The colors of these oxidation products, including red and orange [30], resulted in a reddening of the leaf edges. Apart from the redness at the leaf edges, there were other color changes in the tea leaves. With the loss of moisture and the progression of oxidation reactions, the color of the leaves gradually darkened from the initial emerald green to a dark green, and some even exhibited partial reddish-brown tones. Furthermore, as the fermentation time increased, there were noticeable changes in the texture features of the tea leaves. As shown in Figure 3c, both m and v exhibited an overall decreased trend, indicating that the surface of the tea leaves transitioned from an initially clear texture to a relatively uniform smoothness. On the other hand, s and k show an overall increased trend, signifying an increase in the color hue and an overall darkening of the color.

### 3.3. VIS-NIR Spectral Feature Screening Results

#### 3.3.1. VIS-NIR Analysis

The methodology, as outlined in Section 2.4, involved grouping the sample spectra and calculating the average spectral values for each group. The average spectral curves of the near-infrared spectra were obtained. These curves showed a consistent trend of vibration at the same wavelength band, and changes in the amplitude were attributed to the changes in chemical composition and surface color during tea fermentation affecting the VIS-NIR spectral response characteristics of the samples [31]. Spectral images in the 400–950 nm range revealed prominent absorption peaks at 450, 690, 770, and 900 nm. Among them, the wavelength near 450 nm was related to some organic matter, such as phenolic compounds; the wavelength near 690 nm corresponded to the absorption peak of plant pigments; the absorption peaks near 770 and 900 nm were related to the third overtone of OH [32] and the water content; and the water molecules showed a strong absorption of light at the 770 and 900 nm wavelengths.

#### 3.3.2. Feature Wavelength Screening

To reduce the spectral data dimensions and collinearity, four algorithms—CARS, SPA, VCPA, and VCPA-IRIV—were employed to extract representative feature wavelengths and diminish extraneous information in the data. This facilitated modeling with feature wavelengths, minimized the computational costs, and established a reliable discriminant analysis model. Among these methods, CARS utilizes random sampling strategies [33]. Consequently, based on the obtained statistical results, this method underwent 20 evaluations to assess its reproducibility and stability.

According to the standard normal variate transformation (SNV) and the data smoothed using the Savitzky–Golay (SG) method, the numbers of feature wavelengths obtained using the different extraction algorithms were 51, 12, 11, and 41. After selection, a maximum of 51 feature variables were retained, significantly reducing redundant information in the data, thereby reducing the dimensionality and collinearity of the spectral data. Figure 4 illustrates the distribution of the extracted feature variables across the entire spectrum.

### 3.4. Results of Judging the Fermentation Degree of Oolong Tea

#### 3.4.1. Classification Models and Dataset Partitioning

Classification models typically use training to build a set of use cases for establishing relationships between inputs and outputs. The model is then applied to a test set to assess its classification ability. In this case, the dataset consisting of 315 samples was randomly divided into three groups. Out of these, two groups were designated as training-set samples, comprising a total of 210 samples. The samples in the remaining group were allocated as prediction-set samples, totaling 105 samples. This partitioning strategy ensured a balanced distribution of samples for training and prediction purposes, promoting the robustness and generalization capability of the model. The utilization of randomization in the sample allocation process added an element of statistical rigor to the dataset-splitting methodology. The resulting configuration allowed for effective model training on a substantial subset of the data while maintaining an independent set for rigorous model evaluation and validation.

#### 3.4.2. Single-Sensor Model Analysis

The LDA algorithm was employed for the classification and recognition of both near-infrared spectroscopy (VIS-NIRS) data and image data. Specifically, three types of LDA models were constructed: a VIS-NIRS full-spectrum LDA model, the VIS-NIRS LDA model after applying four variable-selection algorithms, and an image feature LDA model. These models were utilized to investigate the accuracy of detecting the degree of fermentation in Oolong tea using a single sensor. The calibration and prediction results of the six LDA models are presented in Table 1. When employing VIS-NIRS technology alone for the detection of Oolong tea fermentation, spectral bands selected via the VCPA algorithm yielded the most favorable predictive performance, achieving a recognition rate of 71.30% on the prediction set. And employing image-processing techniques alone resulted in a recognition rate of 66.25% on the prediction set. Notably, when assessing the degree of fermentation in Oolong tea, the detection accuracy achieved using VIS-NIRS technology alone surpassed that achieved using image-processing techniques alone. This discrepancy may be attributed to the fact that the internal compositional changes during the fermentation process of Oolong tea are more pronounced than external feature alterations.

#### 3.4.3. Applying Three Traditional Machine Learning Algorithms on the Fusion Data Modeling Results

The three traditional machine learning algorithms were employed for classification and recognition of the fused data, with the results being presented in Table 2. It is observed that the LDA models established based on the four fused datasets exhibit identical performance in both the calibration and prediction sets. However, in the SVM and BPANN models, the fused data incorporating VCPA-selected variables and image features demonstrated the optimal performance in the prediction set. Consequently, we opted for the fused data after VCPA selection for subsequent investigations.

After classifying and modeling the fusion data after VCPA screening, the results achieved with the LDA indicated that the identification rates of the calibration and prediction sets for the three fermentation stages were 85.71% and 80.00% (Table 3), with PCs = 1 (Figure 5a). The first- and second-stage samples were identified with the majority of errors (Table 4). The sensitivity and specificity results (Table 3) confirmed the worse performance of this model, with values ranging in the prediction set from 0.5 to 0.85 and from 0.67 to 0.84, respectively.

The optimized SVM model achieved the peak identification rate in Figure 5b. For the calibration and prediction sets, the SVM model demonstrated robust identification rates, attaining 98.97% and 97.14% (Table 3), respectively. To enhance parameter interaction validation, the logarithm of both c and g was employed as the coordinate axis. Notably, the highest identification rate was observed with specific parameter values, namely c = 45.25 and g = 0.03. The SVM model exhibited superior performance, notably manifesting minimal classification errors across the three stages (Table 4). The sensitivity and specificity values with the prediction set (Table 3) verified the improved performance of this model compared to that of the LDA models, particularly for the first and second stages. 

In Figure 5c, the identification rates of the BPANN model in the calibration and prediction sets are presented. The BPANN model was determined as the optimal configuration for identifying Oolong tea fermentation stages, with results for the calibration and prediction sets of 100% and 94.29% (Table 3), respectively. This model demonstrated a five-epoch structure in the hidden layer (Figure 5d). Furthermore, one validation check was conducted during the training process (Table 4). The sensitivity and specificity of the prediction-set results (Table 3) were similar to those of the SVM model.

#### 3.4.4. Results of Deep Learning Modeling

For the classification and recognition of Oolong tea fermentation levels, both MLP and CNN architectures were employed. In the CNN model, given the limited dataset to prevent overfitting, convolutional kernels of size 3 were employed. The hyperbolic tangent (tanh) activation function was utilized to enhance the model’s ability to fit non-linearities. For the loss function, due to the multi-classification nature of the task, the Cross-Entropy function was chosen. Categorical Cross-Entropy is a loss function commonly used in machine learning and deep learning for multi-classification tasks, playing a pivotal role in neural network training. The prediction set achieved an accuracy of 95.15%. The confusion matrices and loss functions used during the training process are shown in Figure 6a and Figure 6b, respectively.

In the MLP model, considering the relatively small sample size, a three-layer perceptron architecture was chosen for data categorization. ReLU and Softmax were adopted as activation functions to enhance the computational efficiency and interpretability of the model outputs. Similarly, the Cross-Entropy function was selected as the loss function. Additionally, Layer Normalization was applied before each layer’s input to expedite network convergence and improve its generalization capability. The prediction set achieved an accuracy of 91.4%. The confusion matrices and loss functions used during this training process are shown in Figure 6c and Figure 6d, respectively.

### 3.5. Discussion

This study integrated VIS-NIR and image data to establish a classification recognition model for determining the fermentation degree of Oolong tea. The results demonstrate that among the three conventional machine learning models used, the SVM model has a recognition accuracy of 97.14%, which exceeds those of the LDA and BPANN models. In the realm of deep learning models, the CNN model exhibits a recognition accuracy of 95.15%, outperforming the MLP model. This discrepancy in performance could potentially be attributed to the stringent initial parameter requirements of the MLP model, which, if improperly initialized, could lead to training failures or instances of overfitting [34]. Furthermore, deep learning models, overall, demonstrate inferior performance compared to conventional machine learning models, potentially due to the limited volume of available data.

## 4. Conclusions

In this study, an innovative online monitoring system for assessing the Oolong tea fermentation process was developed by combining near-infrared spectroscopy and image-processing techniques. This research establishes a recognition model for determining the fermentation degree of Oolong tea based on data preprocessing, feature wavelength selection, and data fusion. In comparison to the raw spectra, the selected feature wavelengths significantly reduce the redundant information in the data, decreasing the dimensionality and collinearity of the spectral data. Among three traditional machine learning models and two deep learning models, the SVM model exhibits the highest recognition accuracy, reaching 97.14%. It is important to note that this study only explores the identification of fermentation degrees for a specific season and variety of Oolong tea. Future work should involve additional batches of data to enrich the model and enhance its applicability across a broader range of scenarios. The experimental results of this study can guide the development of a non-destructive detection hardware system for Oolong tea fermentation. This system integrates near-infrared spectroscopy and image acquisition devices, contributing to its potential application as a smart tool for processing tea.

## Figures and Tables

**Figure 1 foods-13-01708-f001:**
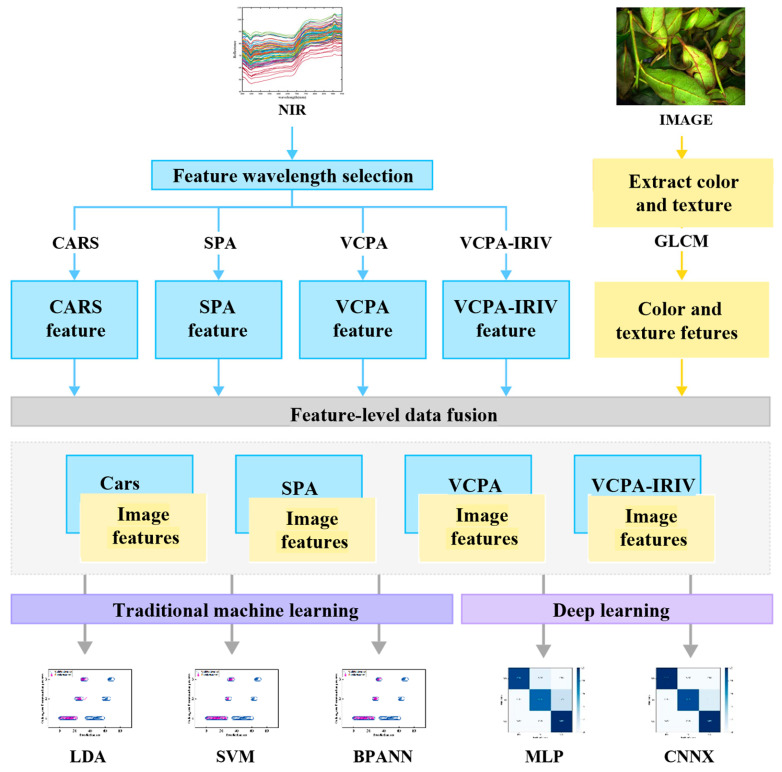
The process diagram of Oolong tea fermentation degree determination based on VIS-NIR and image-processing data fusion technology.

**Figure 2 foods-13-01708-f002:**
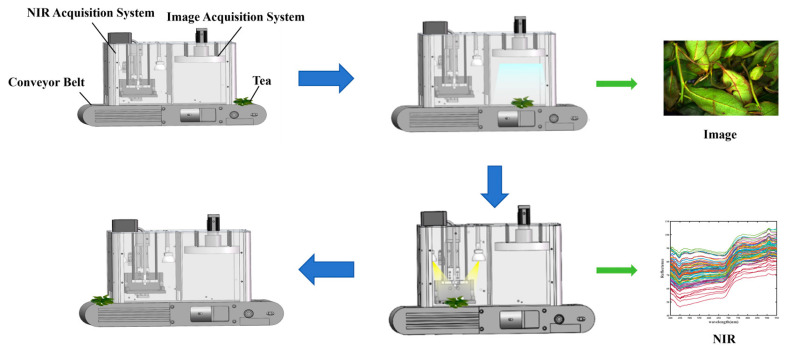
Sampling flow diagram.

**Figure 3 foods-13-01708-f003:**
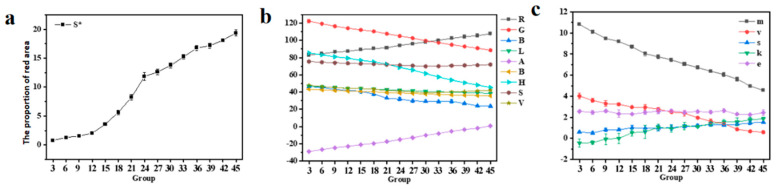
(**a**) Changes in the area of the red part of the leaf; (**b**) RGB, HSV, and Lab changing trends; (**c**) varying trends of texture feature values extracted from the gray co—occurrence matrix.

**Figure 4 foods-13-01708-f004:**
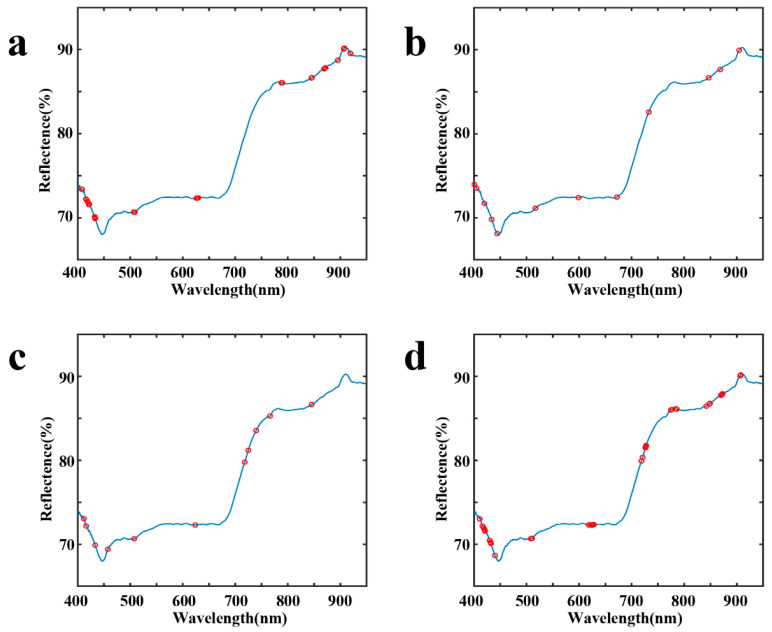
The distribution of the extracted characteristic variables throughout the spectrum: (**a**) CARS; (**b**) SPA; (**c**) VCPA; (**d**) VCPA-IRIV.

**Figure 5 foods-13-01708-f005:**
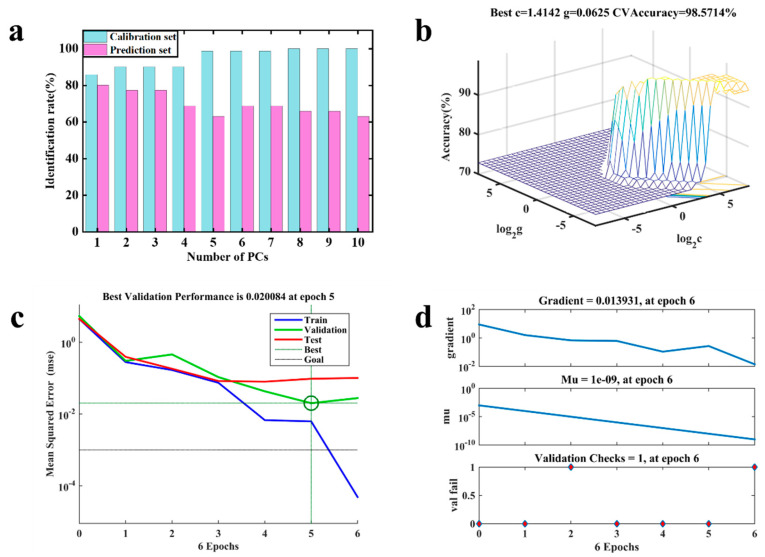
Diagrams of the three traditional machine learning modeling processes: (**a**) LDA, (**b**) SVM, (**c**,**d**) BPANN.

**Figure 6 foods-13-01708-f006:**
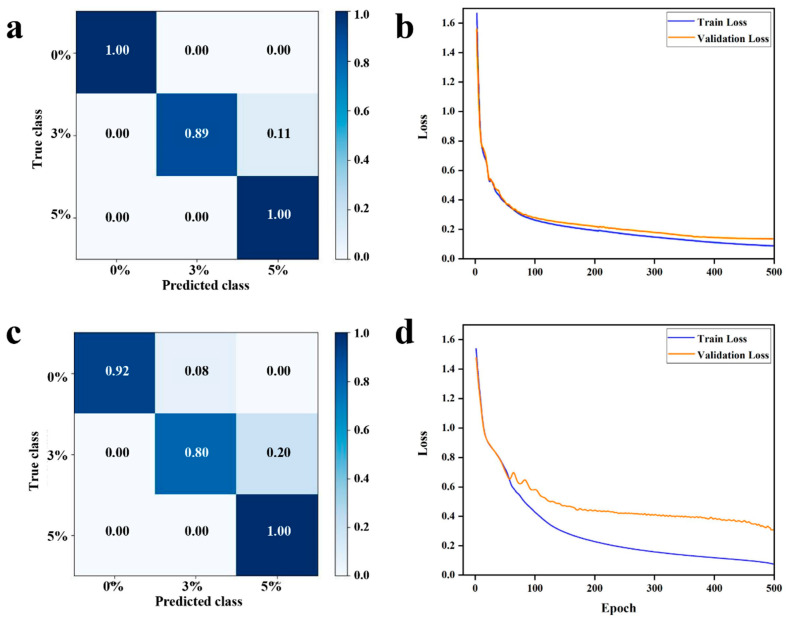
(**a**) Confusion matrix of the CNN model, (**b**) loss function of the CNN model, (**c**) confusion matrix of the MLP model, (**d**) loss function of the MLP model.

**Table 1 foods-13-01708-t001:** Results of the classification and recognition of near-infrared spectral and image data by selecting the optimal PCs in LDA.

Data	VIS-NIR RAW	VIS-NIR CARS	VIS-NIR VCPA	VIS-NIR VCPA-IRIV	VIS-NIR SPA	IMAGE
Calibration set	0.6356	0.6850	0.7378	0.7465	0.6457	0.7034
Prediction set	0.6198	0.6442	0.7130	0.7006	0.6286	0.6625

**Table 2 foods-13-01708-t002:** Results of using three traditional machine learning methods to classify and identify the results of the fused data.

Method	LDA		SVM		BPANN	
	Calibration Set	Prediction Set	Calibration Set	Prediction Set	Calibration Set	Prediction Set
CARS	0.857	0.800	1.000	0.924	0.971	0.857
SPA	0.857	0.800	0.990	0.971	1.000	0.943
VCPA	0.857	0.800	1.000	0.943	1.000	0.971
VCPA-IRIV	0.857	0.800	1.000	0.971	0.967	0.930

**Table 3 foods-13-01708-t003:** Recognition results achieved with the three traditional machine learning classification methods.

Grade	Index	LDA		SVM		BPANN	
		Calibration Set	Prediction Set	Calibration Set	Prediction Set	Calibration Set	Prediction Set
1	Sensitivity	0.8431	0.8462	1.0000	1.0000	1.0000	0.9615
	Specificity	0.8947	0.6667	0.9474	0.8889	1.0000	0.8889
	Accuracy	0.8571	0.8000	0.9857	0.9714	1.0000	0.9429
	Error	0.1143	0.1714	0.0000	0.0000	0.0000	0.0571
2	Sensitivity	0.8000	0.5000	0.9000	0.7500	1.0000	0.7500
	Specificity	0.8667	0.8387	1.0000	1.0000	1.0000	0.9677
	Accuracy	0.8571	0.8000	0.9857	0.9714	1.0000	0.9429
	Error	0.1429	0.8857	0.0143	0.0286	0.0000	0.0571
3	Sensitivity	1.0000	0.8000	1.0000	1.0000	1.0000	1.0000
	Specificity	0.8361	0.8000	0.9836	0.9667	1.0000	0.9333
	Accuracy	0.8571	0.8000	0.9857	0.9714	1.0000	0.9429
	Error	0.0286	0.8286	0.0143	0.0286	0.0000	0.0000

**Table 4 foods-13-01708-t004:** Cross matrix for the identification results achieved with the models in the calibration and prediction sets.

Models	Grade	NC ^a^	Results in the Calibration Set				NP ^b^	Results in the Prediction Set			
			1	2	3	CR ^c^		1	2	3	CR ^c^
LDA	1	153	129	24	0	85.71%	78	66	12	0	80.00%
	2	30	0	24	6		12	0	6	6	
	3	27	0	0	27		15	0	3	12	
SVM	1	153	153	0	0	98.57%	78	78	0	0	97.14%
	2	30	0	27	3		12	0	9	3	
	3	27	0	0	27		15	0	0	15	
BPANN	1	51	51	0	0	100.00%	78	75	3	0	94.29%
	2	10	0	10	0		12	3	9	0	
	3	9	0	0	9		15	0	0	15	

^a^ NC: the number of samples in the calibration set. ^b^ NP: the number of samples in the prediction set. ^c^ CR: correct identification rate.

## Data Availability

The original contributions presented in the study are included in the article, further inquiries can be directed to the corresponding author.

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
