# Peer review of "Online System for Monitoring the Degree of Fermentation of Oolong Tea Using Integrated Visible–Near-Infrared Spectroscopy and Image-Processing Technologies"

_foods, 2024, doi:10.3390/foods13111708_

Round 1
Reviewer 1 Report
Comments and Suggestions for Authors
The authors describe an interesting application to inline monitor the fermentation degree of Oolong tea during the manufacturing process.
The following revisions might be considered:
- did you try to use the techniques separately? The effort of datafusion of two techniques looks very large and expensive. NIR alone might be suitable for the qualitative research question?
- line 75: please provide the botanical name of the variety
- line 90: is this a law? provide reference
- Line 91: how does sampling every 10 minutes over 6.5 hours lead only to 7 samples
- Line 92/93: it is unclear what are batches and how the resulting samples numbers are occurring
- line 99: provide references for the practices of "different researchers"
- section 2: please add a section about software suppliers, version and algorithm references if needed. Please also add a section about method validation approach (this is currently stated in the results, please move and merge)
- line 257 and throughout: please provide city, country for all suppliers
- line 339: why was only 1 PC used. This is highly unusual for PCA/LDA
- Table 1: what are the values in the table. Provide a descriptive legend
- Table 2: perhaps specify in the methods section how you calculate specificity, accuracy and error
- Bottom matter: please provide statement about data availability according to journal guideline. If possible, provide dataset as supplement.
Comments on the Quality of English Language
- the English is non-grammatical on instances and very difficult to understand
- the intext citations must be numbers
- line 116: two lamps
- line 280: darkened of the color?
- line 351: was achieve?
Author Response
Response to Reviewer 1 Comments |
||
1. Summary |
|
|
Thank you very much for taking the time to review this manuscript. Please find the detailed responses below and the corresponding revisions/corrections highlighted/in track changes in the re-submitted files. |
||
2. Questions for General Evaluation |
Reviewer’s Evaluation |
Response and Revisions |
Does the introduction provide sufficient background and include all relevant references? |
Yes/Can be improved/Must be improved/Not applicable |
|
Are all the cited references relevant to the research? |
Yes/Can be improved/Must be improved/Not applicable |
|
Is the research design appropriate? |
Yes/Can be improved/Must be improved/Not applicable |
|
Are the methods adequately described? |
Yes/Can be improved/Must be improved/Not applicable |
|
Are the results clearly presented? |
Yes/Can be improved/Must be improved/Not applicable |
|
Are the conclusions supported by the results? |
Yes/Can be improved/Must be improved/Not applicable |
|
3. Point-by-point response to Comments and Suggestions for Authors |
||
Comments 1: did you try to use the techniques separately? The effort of datafusion of two techniques looks very large and expensive. NIR alone might be suitable for the qualitative research question? |
||
Response 1: Thank you for pointing this out. I have tried to use these two methods separately, and I have added the results of using them separately in lines 346-364 of the revised manuscript |
||
Comments 2: line 75: please provide the botanical name of the variety |
||
Response 2: Iron Goddess Tea. Tieguanyin, literally meaning "iron goddess" in Mandarin Chinese, is a particular and popular variety of Chinese tea. |
||
Comments 3: line 90: is this a law? provide reference |
||
Response 3: Thank you for pointing this out. This study divides the degree of fermentation according to this tradition. |
||
Comments 4: Line 91: how does sampling every 10 minutes over 6.5 hours lead only to 7 samples |
||
Response 4: Here it has been modified to: Sample collection every 10 minutes during fermentation, with 7 samples collected at a time. Line 91 |
||
Comments 5: Line 92/93: it is unclear what are batches and how the resulting samples numbers are occurring |
||
Response 5: I have made corrections here, on lines 91-93 of the revised manuscript. |
||
Comments 6: line 99: provide references for the practices of "different researchers" |
||
Response 6: This is to explain the manual error, the revised content in the revised draft 99-101lines. |
||
Comments 7: section 2: please add a section about software suppliers, version and algorithm references if needed. Please also add a section about method validation approach (this is currently stated in the results, please move and merge) |
||
Response 7:Agree. I added the validation method section and attached the formula to the revised manuscript on lines 246-264 |
||
Comments 8: line 257 and throughout: please provide city, country for all suppliers |
||
Response 8: Thank you, I have finished adding .line 277 |
||
Comments 9: line 339: why was only 1 PC used. This is highly unusual for PCA/LDA |
||
Response9:According to the actual data analysis, the first principal component accounts for 96% of the variable feature expression in the LDA, which means that it contains almost all the discriminant information in the data. In this case, it is reasonable to choose a principal component as a discriminating feature because this component is already strong enough to delineate the major differences between the categories. In addition, the adoption of a single principal component can significantly simplify the complexity of the model, reduce computational costs, and improve the interpretability of the model. Although the data involves three categories, two discriminant components may normally be required to describe a complete separation between categories, but since the first principal component already contains the vast majority of the category-distinguishing information, it is sufficient for efficient classification. In a multi-class problem, reliance on a single principal component can lead to the risk of oversimplifying the data structure. A single component may not fully capture the complex relationships between all categories, especially when the differences between the categories are more than just in the most dominant direction. As a linear technique, LDA is mainly suitable for cases where data types can be clearly separated by linear boundaries. In practice, if the distribution of the data is non-linear, LDA may not be able to effectively distinguish between all categories. Therefore, further use of nonlinear pattern recognition analysis methods to process data, such as support vector machines (SVM), BPANN, etc., can better deal with nonlinear relationships and complex patterns in data, and can more deeply mine data characteristics to capture details that simple LDA may ignore. Therefore, the generalization ability and classification accuracy of the model on diverse data are improved. Table S2 is Eigenvalue and cumulate contribution rates of the principal components. |
||
Comments 10: Table 1: what are the values in the table. Provide a descriptive legend |
||
Response10: Thank you. I've finished it |
||
Comments11: Table 2: perhaps specify in the methods section how you calculate specificity, accuracy and error |
||
Response 11: Finished. Line261-264 |
||
4. Response to Comments on the Quality of English Language |
||
Point 1:line 116: two lamps |
||
Response 1: line 117 (in red) |
||
Point 2: line 280: darkened of the color? |
||
Response 2: line 300 (in red) |
||
Point 3:line 351: was achieve? |
||
Response 3: line 387 (in red) |
||
5. Additional clarifications |
||
|
Table S2. Eigenvalue and cumulate contribution rates of the principal components.
Principal Component |
Eigenvalue |
Percentage of Variance (%) |
Cumulative (%) |
1st |
10.590 |
96.281 |
96.281 |
2nd |
0.347 |
3.155 |
99.437 |
3rd |
0.038 |
0.347 |
99.784 |
4th |
0.013 |
0.126 |
99.910 |
5th |
0.005 |
0.051 |
99.962 |
6th |
0.002 |
0.025 |
99.987 |
7th |
6.96298E-4 |
0.006 |
99.993 |
8th |
4.04615E-4 |
0.003 |
99.997 |
9th |
2.13934E-4 |
0.002 |
99.999 |
10th |
5.27436E-5 |
4.79488E-4 |
99.999 |

Reviewer 2 Report
Comments and Suggestions for Authors
The paper entitled “Online monitoring the degree of fermentation of Oolong tea using integrating near infrared spectroscopy and image processing technologies” presents an interesting study to monitor the fermentation process of Oolong tea by merging data obtained from image capture and visible NIR spectra. The tools used to process the data and evaluate the results were adequate, with the sole exception of the tool used to divide the 315 samples into the different sample sets. This was done randomly, but this is not the best way to do it, as there are different algorithms specialised in this division of samples that work better than the random method. Here are some special remarks to the authors.
1.-The title should be changed as the spectral band it uses is visible wavelengths.
2.- Line 82: How much of the sample was added to the fermenter?
3.- Line 90: Law or tradition?. Replace law by tradition.
4.- Lines 90-93: Not understood. Describe better.
5.- Line 107: Does it fall directly from the fermentation chamber? How do the fermentation chamber and the carrier belt communicate?
6.- Line 149: The range 400-950 nm, includes the visible.
7.- Line 151: Include vis-NIR
8.- Line 151: If the range from 400 to 950 has 550 wavelengths, where does the data of 1517 wavelengths come from? What is the resolution of the equipment?
9.-Lines 209-213: The reference is not included at the end of the document. It is indicated twice in the same paragraph. Indicate only once.
10.- Paragraph 3.1: The conveyor belt is still subjected to a cleaning process between each tea sample?
11.- Lines 292-293: These three wavelengths correspond to the visible. The 450 to the blue, 690 and 770 to the visible red. Only 900 corresponds to the near infrared. The title of the article should be changed as it is misleading.
The article will be ready for publication, once the necessary changes have been made.
Author Response
Response to Reviewer 2 Comments |
||
1. Summary |
|
|
Thank you very much for taking the time to review this manuscript. Please find the detailed responses below and the corresponding revisions/corrections highlighted/in track changes in the re-submitted files. |
||
2. Questions for General Evaluation |
Reviewer’s Evaluation |
Response and Revisions |
Does the introduction provide sufficient background and include all relevant references? |
Yes/Can be improved/Must be improved/Not applicable |
|
Are all the cited references relevant to the research? |
Yes/Can be improved/Must be improved/Not applicable |
|
Is the research design appropriate? |
Yes/Can be improved/Must be improved/Not applicable |
|
Are the methods adequately described? |
Yes/Can be improved/Must be improved/Not applicable |
|
Are the results clearly presented? |
Yes/Can be improved/Must be improved/Not applicable |
|
Are the conclusions supported by the results? |
Yes/Can be improved/Must be improved/Not applicable |
|
3. Point-by-point response to Comments and Suggestions for Authors |
||
Comments 1: The title should be changed as the spectral band it uses is visible wavelengths. |
||
Response 1: Thank you for pointing out this problem, I have revised the title |
||
Comments 2: Line 82: How much of the sample was added to the fermenter? |
||
Response 2: I added the sample weight to the revised manuscript. Line79 |
||
Comments 3: Line 90: Law or tradition?. Replace law by tradition. |
||
Response 3: Thank you for pointing this out. I have replaced law with tradition.line90 |
||
Comments 4: Lines 90-93: Not understood. Describe better. |
||
Response 4: I have modified this part of the statement. Line88-93
|
||
Comments 5: Line 107: Does it fall directly from the fermentation chamber? How do the fermentation chamber and the carrier belt communicate? |
||
Response 5: The conveyor belt is installed under the fermentation chamber, and the tea leaves can cover the conveyor belt. |
||
Comments 6: Line 149: The range 400-950 nm, includes the visible. |
||
Response 6:Thank you, I have replaced all NIR with vis-NIR |
||
Comments 7: Line 151: Include vis-NIR |
||
Response 7:Thank you, I have replaced all NIR with vis-NIR |
||
Comments 8: Line 151: If the range from 400 to 950 has 550 wavelengths, where does the data of 1517 wavelengths come from? What is the resolution of the equipment? |
||
Response 8: Spectral data was collected in the wavelength range of 400-950 nm at a spectral interval of approximately 0.42, containing 1517 variables. |
||
Comments 9: Lines 209-213: The reference is not included at the end of the document. It is indicated twice in the same paragraph. Indicate only once. |
||
Response9: I've added this quote. |
||
Comments 10: Paragraph 3.1: The conveyor belt is still subjected to a cleaning process between each tea sample? |
||
Response10: Since the tea will completely cover the conveyor belt, the cleanliness of the conveyor belt will not affect the collection of data, so there is no need to clean. |
||
Comments11: Lines 292-293: These three wavelengths correspond to the visible. The 450 to the blue, 690 and 770 to the visible red. Only 900 corresponds to the near infrared. The title of the article should be changed as it is misleading. |
||
Response 11: Thank you for pointing out this problem, I have revised the title |
